# Role of the Exosome in Ovarian Cancer Progression and Its Potential as a Therapeutic Target

**DOI:** 10.3390/cancers11081147

**Published:** 2019-08-10

**Authors:** Koji Nakamura, Kenjiro Sawada, Masaki Kobayashi, Mayuko Miyamoto, Aasa Shimizu, Misa Yamamoto, Yasuto Kinose, Tadashi Kimura

**Affiliations:** 1Department of Obstetrics and Gynecology, Graduate School of Medicine, Osaka University, 2-2, Yamadaoka, Suita, Osaka 5650871, Japan; 2Department of Molecular Oncology, H. Lee Moffitt Cancer Center & Research Institute, Tampa, FL 33612, USA; 3Penn Ovarian Cancer Research Center, Perelman School of Medicine, University of Pennsylvania, Biomedical Research Building II/III, 421 Curie Blvd, Philadelphia, PA 19104, USA

**Keywords:** exosome, ovarian cancer, peritoneal dissemination

## Abstract

Peritoneal dissemination is a distinct form of metastasis in ovarian cancer that precedes hematogenic or lymphatic metastasis. Exosomes are extracellular vesicles of 30–150 nm in diameter secreted by different cell types and internalized by target cells. There is emerging evidence that exosomes facilitate the peritoneal dissemination of ovarian cancer by mediating intercellular communication between cancer cells and the tumor microenvironment through the transfer of nucleic acids, proteins, and lipids. Furthermore, therapeutic applications of exosomes as drug cargo delivery are attracting research interest because exosomes are stabilized in circulation. This review highlights the functions of exosomes in each process of the peritoneal dissemination of ovarian cancer and discusses their potential for cancer therapeutics.

## 1. Introduction

Ovarian cancer is the most lethal human gynecological malignancy [1,2]. When it is diagnosed at an early stage, the five-year relative survival rate is over 90%. However, more than two-thirds of ovarian cancer patients are diagnosed at advanced stages with peritoneal dissemination, and their prognoses are poor in spite of novel molecular targeted therapies which were developed more than 20 years ago [2]. Therefore, understanding the underlying mechanism of the peritoneal dissemination of ovarian cancer is essential to overcome and control this distinct form of metastasis.

Extracellular vesicles (EVs) have become the focus of increase research interest due to their diverse functions in physiology and pathology [3]. EVs are classified into various subtypes such as exosomes, microvesicles, and apoptotic bodies according to their size and origin. Exosomes are 30–150 nm-sized vesicles of endocytic origin which play key roles in cancer biology by mediating cell-to-cell communication through the transfer of proteins, nucleic acids, and lipids [4,5]. Exosome research has rapidly expanded over the last decade and has shown their immense potential as promising diagnostic and therapeutic biomarkers, as well as therapeutic targets in various types of cancers [6,7,8,9]. Indeed, recent studies have demonstrated that exosomes contribute to tumor progression and metastasis by mediating epithelial-to-mesenchymal transition (EMT), migration, invasion, angiogenesis, immune modulation and metabolic, epigenetic and stromal reprogramming to a cancer-associated phenotype in a pre-metastatic niche [10,11]. In ovarian cancer research, exosomes promote peritoneal dissemination through the interaction between cancer cells and their microenvironments [12,13,14]. In this review, we highlight the latest findings of the mechanisms of ovarian cancer peritoneal dissemination facilitated by exosomes and discuss the implications of exosomes as future therapeutic targets.

## 2. The Role of Exosomes in Ovarian Cancer Peritoneal Dissemination

During the process of peritoneal dissemination, ovarian cancer cells detach from the primary site of origin (ovary and/or fallopian tube). Subsequently, these cancer cells spread into the peritoneal cavity and attach to the surface of peritoneal organs, most notably the omentum, which is the predominant site of ovarian cancer metastasis [15]. All of the organs within the peritoneal cavity are lined with a single layer of mesothelial cells that cover an underlying stroma composed of an extracellular matrix and stromal cells [16]. Once ovarian cancer cells attach to this layer, they invade through the mesothelial cell barrier into the peritoneum, omentum, and bowel serosa. The overall process of ovarian cancer peritoneal dissemination is summarized in Figure 1.

Several studies have shown that cancer cell-derived exosomes reprogram or educate other cells to support tumor survival and promote metastasis [17,18]. However, exosomes secreted by cells from the tumor microenvironments including fibroblasts, mesothelial cells, adipocytes, and immune cells also affect cancer cells [19]. It is now believed that the exosome-mediated crosstalk between cancer cells and the tumor microenvironment is deeply involved in each step of peritoneal dissemination. In this chapter, we describe the latest findings of the interaction between cancer cells and tumor microenvironments that are facilitated by exosomes, focusing on each step during ovarian cancer dissemination (Table 1).

### 2.1. Shedding of Cancer Cells from Primary Tumor Sites

The initial step of peritoneal dissemination is the shedding of cancer cells from their primary site throughout the peritoneal cavity [43]. The prerequisite for this step is the loss of cell–cell contact between cancer cells, which undergo EMT and display an aggressive phenotype [43]. A number of studies have demonstrated that exosomes from both cancer cells and tumor microenvironments mediate this morphological change [44]. Proteomic studies have shown that exosomes include transforming growth factor beta (TGFβ), tumor necrosis factor alpha (TNFα), interleukin (IL)-6, β-catenin and matrix metalloproteinases (MMPs), and that these are involved in the process of EMT [44]. Mesenchymal stem cells (MSC) and macrophages derived from exosomes promote the migration and/or invasion of breast cancer via the activation of Wnt signaling [45]. TGFβ1 was enriched in ovarian cancer-associated fibroblast (CAF)-derived exosomes compared to normal fibroblasts, and CAF-derived exosomes promoted the EMT of cancer cells through the activation of SMAD signaling, which enhanced the migration and invasion ability of these cancer cells [42]. Exosomal miRNAs have been implicated in the regulation of EMT-related pathways [44]. Stromal fibroblast-derived miR-409 induced EMT as well as the stemness of prostate cancer cells [46].

### 2.2. Floating in the Peritoneal Cavity

Once cancer cells shed into the peritoneal cavity, they face several challenges for survival, including anoikis and immune surveillance [47]. In this step, exosome-orchestrated cell–cell interactions between cancer cells and the tumor microenvironment support cells that attach cancer cells to the peritoneal surface and subsequently form metastatic nodules [48]. Gutwein et al. showed that soluble L1 adhesion molecule (CD171)-containing exosomes are present in the ascites of ovarian cancer patients using sucrose density gradient centrifugation. They demonstrated that L1-containing exosomes trigger cell migration and extracellular signal-regulated kinase (ERK) phosphorylation in vitro [35].

#### 2.2.1. Hypoxia-Induced Exosomes Promote Cancer Cell Survival

During peritoneal metastases, cancer cells detach from the primary site, float into the peritoneal cavity without access to vascular supply, and are therefore exposed to hypoxic conditions. Hypoxia causes the cells to acquire a more aggressive malignant phenotype that allows them to metastasize [49]. Recent reports suggest that exosome production is upregulated in a hypoxic microenvironment in a hypoxia-inducible factor (HIF)-1α-dependent or independent manner in various cancer types, including ovarian cancer [31,50,51]. Hypoxia-induced cancer-derived exosomes deliver oncogenic signals to both tumor cells and the tumor environment. They play pivotal roles in proliferation, invasion, angiogenesis, stemness, resistance to the treatment, and immune evasion [48]. Dorayappan et al. showed that ovarian cancer cells exposed to hypoxia increased their exosome release. Exosomes isolated from hypoxic ovarian cancer cells using the ultracentrifugation method promote metastasis compared with those from normoxia in vivo. They demonstrated that phosphorylated STAT3 is enriched in these exosomes, which promotes cell migration and invasion [31]. Tumor-associated macrophages (TAMs) are one of the most common immune-related stromal cells in the tumor microenvironment, and the communication between cancer cells and TAMs is crucial for the progression of ovarian cancer [52]. Chen et al. demonstrated that hypoxic ovarian cancer-derived exosomes promote macrophage M2 polarization. TAMs educated by hypoxic exosomes promote ovarian cancer cell proliferation and migration. Through the comparison of miRNA profiling of ovarian cancer-derived exosomes in normoxia with those in hypoxia using miRNA microarray, they showed that several miRNAs, such as miR-21-3p, miR-125 b-5p and miR-181 d-5p, were enriched in hypoxic ovarian cancer-derived exosomes They further demonstrated that these three miRNAs regulate the SOCS4/5/STAT3 pathway to modulate macrophage M2 polarization [29]. The same group also showed that ovarian cancer exosomal miR-940 is upregulated in hypoxia and induces macrophage M2 polarization in vitro [30]. More recently, Zhu et al. demonstrated that hypoxia stimulation induces macrophage M2 polarization and enhances the internalization of macrophage-secreted exosomes by ovarian cancer cells, and showed that miR-223 in TAM-derived exosomes reduced the sensitivity of ovarian cancer cells to cisplatin [40].

#### 2.2.2. Exosomes in Malignant Ascites Support Cancer Cell Survival and Premetastatic Niche Formation

Malignant ascites occurs in most patients with high-grade serous ovarian cancer [53]. Studies have shown that malignant ascites serves as an important tumor microenvironment, which is enriched with tumor-promoting factors such as TGFβ, hepatocyte growth factor (HGF), IL-6, IL-8, IL-10, vascular endothelial growth factor (VEGF) and so on. Through these factors, malignant ascites supports ovarian cancer cell proliferation, invasion and anti-apoptosis, and subsequently contributes to chemo-resistance and tumor heterogeneity [54,55]. A variety of studies have investigated the underlying mechanism of malignant ascites-derived exosomes in ovarian cancer progression. Runz et al. collected malignant ascites-derived exosomes from 16 ovarian cancer patients and found that these exosomes contain CD24 and Epithelial Cell Adhesion Molecule (EpCAM), which are known as stem cell markers and promote cancer invasion [34]. Shender et al. conducted the proteome–metabolome profiling of ovarian cancer ascites and identified a number of proteins and metabolites that are involved in key cancer signal transduction in malignant ascites-derived exosomes [56]. Gutwein et al. demonstrated that soluble L1 (CD171) is contained in vesicles from malignant ascites in ovarian cancer patients and serves as a potent inducer of cancer cell migration [35]. Malignant ascites-derived exosomes also contribute to forming a premetastatic niche. Graves et al. showed that malignant ascites-derived exosomes contain gelatin lysis enzymes including MMP-2, MMP-9, and urokinase-type plasminogen activator (uPA), and these proteolytic enzymes in exosomes regulate cell invasion [36]. Clancy et al. demonstrated that exosomes isolated from malignant ascites of ovarian cancer patients can deliver membrane-type 1 MMP (MT1-MMP) to the cell surface, which promotes ovarian cancer invasion [37]. These studies suggest the importance of exosomes in extracellular matrix degradation, a key process of peritoneal dissemination.

### 2.3. Cancer Cell Attachment to the Peritoneal Cavity

All of the organs within the peritoneal cavity are lined with a single layer of peritoneal mesothelial cells (PMCs) that cover an underlying stroma composed of ECMs and stromal cells. During ovarian cancer dissemination, once ovarian cancer cells attach to this layer, they invade through the barrier of PMCs into the peritoneum, omentum, and bowel serosa [57,58]. Under normal conditions, PMCs act as a mechanical barrier that protects intra-abdominal organs [12]. However, through the communication between cancer cells via exosomes, PMCs can also serve as a metastatic niche that promotes cancer cell adhesion and invasion, which are essential steps for the peritoneal dissemination of ovarian cancer [15]. Nakamura et al. described how ovarian cancer-derived exosomes reprogram PMCs with a mesenchymal phenotype (i.e., mesothelial–mesenchymal transition (MMT)) by transferring CD44 and subsequently promote ovarian cancer cell invasion and metastasis [12]. Stimulated mesothelial cells degrade ECMs through MMP-9 secretion, and the mesothelial barrier can be penetrated through this morphological change in PMCs. Yoshimura et al. demonstrated the contribution of exosomal miRNA in reprogramming PMCs. They found that miR-99a-5p is elevated in exosomes in ovarian cancer cell lines compared with those from immortalized normal ovarian epithelium by miRNA microarray. MiR-99a-5p was also elevated in the sera of ovarian cancer patients compared with those of healthy volunteers and patients with benign gynecological tumors. They further demonstrated that exosomal miR-99a-5p promoted cancer cell invasion by upregulating fibronectin and vitronectin expression in PMCs [13]. Yokoi et al. showed that the cancer-derived exosomes efficiently induce apoptotic cell death in PMCs, thus resulting in the destruction of the peritoneal mesothelium barrier [14]. Through whole transcriptome analysis, they identified MMP1 mRNAs as being packaged in exosomes from highly metastatic cancer cells and mediate this apoptotic change. [14]. The role of cancer-derived exosomes in PMCs are summarized in Figure 2. Similarly, gastric cancer-derived exosomes promote MMT in PMCs through the transfer of exosomal miR-21-5p, which activates the TGF/SMAD pathway by targeting SMAD7 and subsequently provides a favorable environment for peritoneal dissemination [59,60]. Deng et al. showed that gastric cancer-derived exosomes elicit injury of the mesothelial barrier through the induction of apoptosis as well as MMT in PMCs [61].

### 2.4. Formation of a Metastatic Tumor

Once cancer cells attach to PMCs that line the surface of the peritoneum, productive reciprocal interactions between the cancer cells and their new microenvironment can occur, which leads to the successful establishment of metastatic tumors [62]. During invasion and metastasis, cancer cells change the normal stroma into a “reactive” environment, which promotes the growth and viability of tumor cells. The key components of the tumor microenvironment are CAFs, TAMs and other immune cells, endothelial cells, adipocytes, pericytes, ECM proteins, etc. [63]. In this section, we describe the exosome-mediated communication between cancer cells and each component of the tumor microenvironment.

#### 2.4.1. Endothelial Cells

Angiogenesis is required for ovarian cancer progression and is an established therapeutic target for advanced diseases. Several large phase III clinical trials have shown the efficacy of vascular endothelial growth factor (VEGF) targeting therapy in ovarian cancer [64,65]. The exosome has emerged as an important mediator in vascular remodeling including angiogenesis in many diseases [66]. Yi et al. showed that high-grade serous ovarian cancer-derived exosomes were transported into primary human umbilical vein endothelial cells (HUVECs) and induced vascular formation [20]. Proteomic profiles suggested that proteins including activating transcription factor 2 (ATF2), metastasis associated 1 (MTA1), rho-associated kinase (ROCK)1/2 in exosomes potentially contribute to angiogenesis. Tang et al. demonstrated that soluble E-cadherin (sE-cad), which is a potent inducer of angiogenesis, is abundantly released via exosomes derived from ovarian cancer and induced angiogenesis [21]. They revealed that sE-cad-positive exosomes heterodimerized with vE-cadherin on endothelial cells and transduced the sequential activation of β-catenin and NFκB signaling. Recent study has focused on the role of an exosomal long non-coding RNA in angiogenesis and showed that the long non-coding RNA, metastasis-associated lung adenocarcinoma transcript 1 (*MALAT1*), in ovarian cancer-derived exosomes can be transferred to HUVECs [22]. MALAT1-containing exosomes stimulated angiogenesis-related gene expression in HUVECs and accordingly promoted angiogenesis.

#### 2.4.2. Fibroblasts (CAFs)

Fibroblasts constitute one of the most abundant cell types in the stroma. They are essential elements for normal tissue homeostasis and function and are generally anti-tumorigenic [67]. A number of recent studies have shown that cancer cells recruit surrounding normal fibroblasts in tumor stroma and reprogram them into CAFs [68,69]. Furthermore, Giusti et al. demonstrated that normal fibroblasts present a CAF-like phenotype by the treatment of ovarian cancer cell-derived exosomes in vitro [23]. CAFs present a pathologically activated phenotype that enables them to influence cancer progression, including dissemination through the remodeling of ECM or by affecting cancer cells or other components of the tumor microenvironment [70,71]. CAF-derived exosomes promote cancer progression including growth, migration, and invasion in multiple cancer types [72,73,74]. Luga et al. reported that fibroblast-secreted exosomes promote breast cancer cell protrusive activity and motility via Wnt-planar cell polarity (PCP) signaling [73]. Khazaei S. et al. revealed the role of exosomal miR-451 in the esophageal stromal tumor microenvironment as a signaling molecule to provide a favorable niche for cancer cell migration and progression [74]. 

#### 2.4.3. Adipocytes

If metastasis is a random event, all organs in contact with peritoneal fluid would have an equal distribution of metastases. However, ovarian cancer preferentially metastasizes to adipose tissue and omentum [75]. Indeed, epidemiological data shows that a higher body mass index (BMI) is associated with adverse survival among ovarian cancer patients [76,77]. Adipocytes are known to contribute to the metastatic cascade in ovarian cancer through the production of adipokines [78,79], and several studies have demonstrated that the mechanism of interaction between cancer cells and adipocytes is mediated by exosomes [80]. Cho et al. showed that exosomes derived from ovarian cancer cells induce the transformation of adipose tissue-derived mesenchymal stem cells (AD-MSCs) with CAF phenotypes [24]. Au Yeung et al. reported that cancer-associated adipocytes-derived exosomal miR-21 is transferred to ovarian cancer cells and suppresses cancer cell apoptosis by targeting apoptotic protease activating factor 1 (APAF1), a key regulator of apoptosis [41]. Thereby, exosomal miR-21 confers paclitaxel resistance [41]. Hoshino et al. showed that cancer cell-derived exosomes that express unique integrins can determine organotropic metastasis by preparing a pre-metastatic niche [81]. Proteomics data revealed that exosomal integrins α6β4 and α6β1 are associated with lung metastasis, while exosomal integrin αvβ5 is linked to liver metastasis, suggesting that ovarian cancer-derived exosomes may play a key role in a metastatic preference for the omentum.

#### 2.4.4. Immune Cells

In general, circulating and disseminated tumor cells acquire a variety of immune-escape mechanisms, including alterations in the expression of major histocompatibility complex (MHC) molecules, Natural killer (NK)-cell ligands, Fas, Fas ligand (FasL), and immune-checkpoint molecules including CD47 and programmed cell death 1 ligand 1 (PD-L1) [82]. There is emerging evidence that suggests exosomes play critical roles in the crosstalk between cancer cells and the immune system during immuno-editing, evasion from immune surveillance, and the formation of metastases by suppressing the activation of T- and NK cells, monocytes, modulating T-cell inhibitory molecule expression, and inducing CD8+ T-cell apoptosis [83,84]. Ovarian cancer-derived exosomes have been shown to reversibly inhibit T-cell activation. Taylor et al. demonstrated that exosomes from ascites of ovarian cancer patients suppress T-cell activation signaling components, CD3-zeta and Janus kinase (JAK) 3, and induce T-cell apoptosis in vitro. They also showed that exosomes express FasL [38]. Czystowska et al. reported that arginase-1 (ARG1)-carrying exosomes accelerated ovarian cancer growth by suppressing T-cells. They found ARG1-carrying exosomes in the ascites and plasma of ovarian cancer patients, and ARG1-containing EVs are transported to draining lymph nodes, taken up by dendritic cells and inhibit antigen-specific T-cell proliferation [25]. Kelleher et al. found that phosphatidylserine expressed on the surface of the cancer-derived exosome inhibited T-cell function and further identified that T-cells pulsed with ovarian cancer ascites-derived exosomes during TCR-dependent activation inhibited several activation endpoints including the translocation of NFκB and NFAT into the nucleus, upregulation of CD69 and CD107a, production of cytokines, and cell proliferation, which all lead to immunosuppression [26,27]. Meng et al. demonstrated that lysophosphatidic acid upregulates FasL on the surface of ovarian cancer-derived exosomes, and thereby FasL -carrying exosomes induced activated T-cell apoptosis and promoted metastasis [28]. Chen et al. demonstrated that stimulation with interferon-γ (IFN-γ) increased the amount of PD-L1 on the metastatic melanoma-derived exosome, which further suppressed the function of CD8 T cells and facilitated tumor growth [85]. Poggio et al. reported that exosomal PD-L1 from several types of cancers can be a major regulator of tumor progression through its suppression of T-cell activation in draining lymph nodes, and that its inhibition leads to long-lasting, systemic anti-tumor immunity [86]. Exosomes derived from ovarian cancer patient ascites are internalized by NK cells and can induce immune suppression [32]. Labani-Motlagh et al. showed that exosomes derived from ovarian cancer cells or malignant ascites inhibited NK cell activity by impairing NKG2D-mediated cytotoxicity in NK cells [33]. Peng et al. demonstrated that FasL and Tumor necrosis factor-related apoptosis-inducing ligand (TRAIL)-containing exosomes isolated from ovarian cancer ascites induced apoptosis in dendritic and peripheral blood mononuclear cells [39].

## 3. Therapeutic Potential of Exosomes in the Peritoneal Dissemination of Ovarian Cancer

Accumulating evidence has demonstrated that exosomes can serve as a therapeutic modality as well as a therapeutic target for cancer treatment [87,88]. In this chapter, we introduce recent advances in the therapeutic potential of exosomes and discuss future perspectives regarding their potential as an ovarian cancer therapy.

### 3.1. Exosomes as Drug Delivery Vehicles

Because exosomes are stable in the circulation and do not induce immune rejection, their therapeutic applications as drug delivery systems have grown into an attractive research area [87,89]. Advances of nanotechnology enable the encapsulation of therapeutic agents such as chemotherapeutic drugs, small molecules, miRNAs, and siRNAs into exosomes [90]. Soo Kim et al. showed that the exosomes carrying paclitaxel into macrophage exosomes increased cytotoxicity more than 50 times to drug-resistant Madin-Darby canine kidney (MDCK) cells and demonstrated nearly complete co-localization between delivered exosomes and cancer cells in a murine Lewis lung carcinoma pulmonary metastasis model, as well as having a potent anticancer effect [91]. Kamerkar et al. engineered exosomes derived from normal fibroblast-like mesenchymal cells to carry siRNA or shRNA specific to oncogenic KRAS^G12D^. Compared to liposomes, engineered exosomes target oncogenic KRAS with an enhanced efficacy that is CD47-dependent and allows circulating monocytes to evade phagocytosis, increasing the half-life in the circulation [92,93]. In ovarian cancer, exosomes derived from adipose mesenchymal stem cells inhibited the cell proliferation of A2780 human ovarian cancer cells by blocking the cell cycle and activating mitochondria-mediated apoptosis signaling, indicating the importance of exosomal miRNAs in this inhibitory pathway [94]. These results suggest that therapeutic exosomes can be applied to peritoneally disseminated ovarian cancer cells. Although there are still many challenges for their clinical use such as large-scale manufacturing, cell sources, and their specificity to target cells, engineering exosomes still remains a promising therapeutic strategy for ovarian cancer treatment.

### 3.2. Exosome-Based Immunotherapy

Immunotherapy is an emerging field in cancer treatment, and exosomes can be developed for cancer immunotherapy because of their immunomodulatory potential [88]. In general, dendritic cells (DCs) play pivotal roles that aim to eliminate cancer cells through T-cell activation in the first steps of the cancer/immunity cycle [95]. DC-derived exosomes have the potential to facilitate immune cell-dependent tumor rejection by activating T-cells and NK cells [96,97,98]. Accordingly, several clinical trials using DC-derived exosomes have been conducted in advanced malignancies to confirm the feasibility and safety of this approach [99]. In a phase II clinical trial, 26 advanced non-small cell lung cancer (NSCLC) patients who did not respond to platinum-based chemotherapy received DC-derived exosomes pulsed with interferon-γ [100]. Among these patients, 22 completed therapy, and seven patients (32%) exhibited stable disease for more than four months. Exosomal PD-L1 can be a major regulator of tumor progression through its ability to suppress T-cell activation, suggesting that targeting both cell-surface and exosome PD-L1 presentation could be a novel therapeutic strategy [85,86]. Despite the lack of evidence for exosome-based ovarian cancer immunotherapy, these findings suggest there is the potential for exosome-based immunotherapies to be used for ovarian cancer.

### 3.3. Exosomes as Therapeutic Target

Given that exosomes play pivotal roles in cancer progression, targeting exosome biogenesis and secretion has potential clinical implications for future cancer therapies. However, there are currently no clinically available drugs that efficiently eliminate deleterious exosomes in cancer patients. Through high-throughput drug screening from a total of 4580 compounds, five potent exosome biogenesis and secretion inhibitors were identified: tipifarnib, neticonazole, climbazole, ketoconazole, and triadimenol [101]. Further studies will be needed to test their in vivo efficacy. Several other drugs have been reported as exosome biogenesis and/or secretion inhibitors: for example, GW4869 is an inhibitor of the ceramide biosynthesis regulator that impairs exosome secretion from 293T cells [102]. Indeed, GW4869 was also shown to inhibit exosome secretion from ovarian cancer cells and ovarian cancer invasion [12]. These findings implicate a potential utility of exosome inhibitors as novel adjunct therapeutic strategies for advanced ovarian cancer. 

## 4. Conclusions

Overcoming peritoneal dissemination remains a primary challenge in treating ovarian cancer. Therefore, it is crucial to elucidate the underlying mechanism of this distinct pattern of metastasis. Emerging evidence has shown that exosomes facilitate the peritoneal dissemination of ovarian cancer in several developmental steps by mediating intercellular communication between cancer cells and the tumor microenvironment. Furthermore, recent studies have implicated the potential of both exosome-targeting and exosome-based therapies. Although there are many challenges to be addressed in the clinical use of exosome therapy, the development of the treatment of the peritoneal dissemination of ovarian cancer is still an attractive research area. 

## Figures and Tables

**Figure 1 cancers-11-01147-f001:**
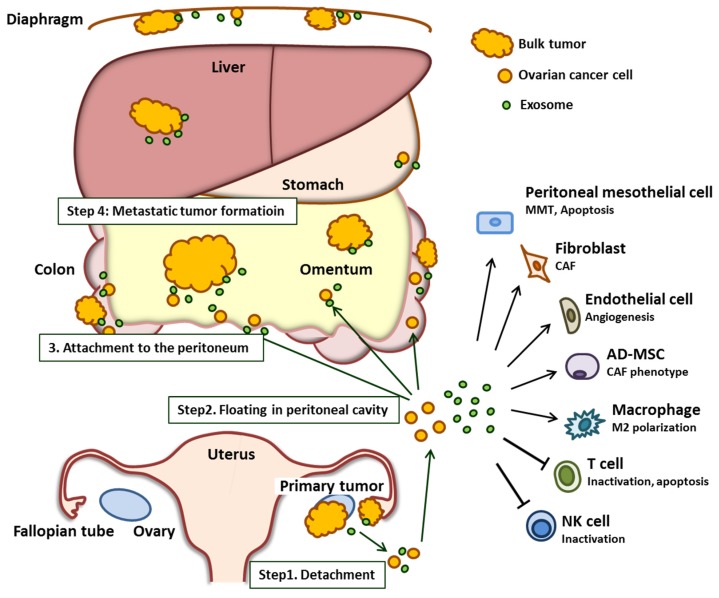
Overview of the role of the ovarian cancer-derived exosome in peritoneal dissemination. Exosomes secreted by the ovarian cancer cell promote each step of peritoneal dissemination by mediating the interaction between the cancer cell and the components of the tumor microenvironment. The stimulated tumor microenvironment supports cancer progression. Abbreviations: MMT; mesothelial–mesenchymal transition, CAF; cancer-associated fibroblast, AD-MSC; adipose tissue-derived mesenchymal stem cell; NK; natural killer.

**Figure 2 cancers-11-01147-f002:**
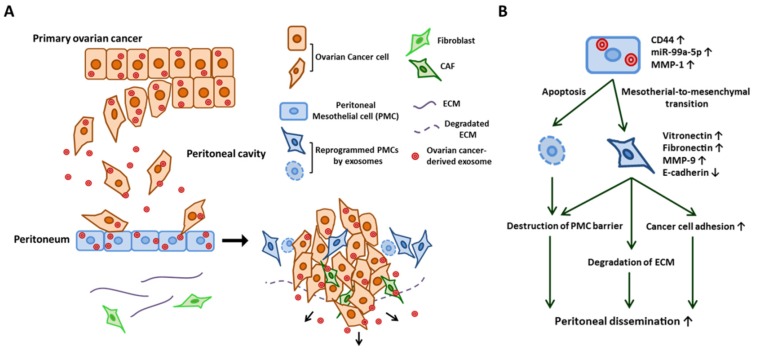
Exosome-mediated reprogramming of peritoneal mesothelial cells (PMCs) promotes the dissemination of ovarian cancer. (**A**) Schematic modeling of early metastasis to the peritoneum. Ovarian cancer-derived exosomes are transferred into PMCs and reprogram PMCs. Exosome-stimulated PMCs undergo mesothelial–mesenchymal transition and then induce the degradation of ECM and upregulate cell adhesion molecules. The exosome also induces the apoptosis of PMCs. MMT and the apoptosis of PMCs destroy the PMC barrier of the peritoneal surface. Cancer-derived exosomes also reprogram the fibroblast into CAFs. (**B**) Proposed mechanism of the reprogramming of PMC by ovarian cancer-secreted exosome (see text for details).

**Table 1 cancers-11-01147-t001:** Intercellular communication mediated by exosomes during ovarian cancer progression.

**A. Exosomes Derived from Cancer Cells**					
	**Method of Exosome Isolation**	**Exosome Content**	**Target Cell**	**Effect on Cancer Progression (Mechanism)**	**Ref**
	Ultracentrifugation	CD44	PMC	Invasion (upregulation of vitronectin/fibronectin in PMC)	[12]
	Ultracentrifugation	miR-99a-5p	PMC	Adhesion/invasion (upregulation of vitronectin/fibronectin in PMC)	[13]
	Ultracentrifugation	MMP-1 mRNA	PMC	Metastasis (apoptosis of PMC)	[14]
	Ultracentrifugation	ATF2, MTA1, ROCK1/2	HUVEC	Angiogenesis	[20]
	Ultracentrifugation	soluble E-cadherin	Endothelial cell	Angiogenesis (β-catenin and NFκB signaling activation in endothelial cell)	[21]
	Kit (Thermo Fisher Scientific)	MALAT1	HUVEC	Angiogenesis	[22]
	Ultracentrifugation		Fibroblast	Proliferation/invasion (activation of fibroblast to CAF)	[23]
	Ultracentrifugation		AD-MSC	(Activation of AD-MSC to CAF-like phenotype)	[24]
	Ultracentrifugation/immunomagnetic beads	ARG-1	T-cell	Immune suppression (inhibition of T cell activity)	[25]
	Ultracentrifugation	Phosphatidylserine	T-cell	Immune suppression (inhibition of T cell activity)	[26,27]
	Ultracentrifugation	Fas ligand	T-cell	Immune suppression (apoptosis of T cell)	[28]
	Kit (Life Technologies)	miR21–3p, miR125b-5p, miR181d-5p (hypoxia induced)	Macrophage	Proliferation/migration (M2 polarization of macrophage)	[29]
	Kit (Life Technologies)	miR-940 (hypoxia induced)	Macrophage	Proliferation/migration (M2 polarization of macrophage)	[30]
**B. Exosomes Derived from Other Sources**					
**Source of Exosome**	**Method of Exosome Isolation**	**Exosome Content**	**Target Cell**	**Effect on Cancer Progression (Mechanism)**	**Ref**
Ascites/cancer cell	Ultracentrifugation	STAT3/Fas (hypoxia induced)	Cancer cell	Migration/invasion/metastasis	[31]
Ascites/cancer cell	Ultracentrifugation	Phosphatidylserine	NK cell	Immune suppression (inhibition of NK cell activity)	[32]
Ascites/cancer cell	Sucrose density fractionation	NKG2D and DNAM-1 ligands	NK cell	Immune suppression (inhibition of NK cell activity)	[33]
Ascites	Ultracentrifugation/sucrose density fractionation	CD24/EpCAM	Cancer cell	Invasion	[34]
Ascites	Sucrose density fractionation	soluble L1 (CD171)	Cancer cell	Migration	[35]
Ascites	Ultracentrifugation	MMP-2, MMP-9, uPA	Cancer cell	Invasion	[36]
Ascites	Ultracentrifugation	MT1-MMP	Cancer cell	Invasion	[37]
Ascites	Ultracentrifugation		T-cell	Immune suppression (apoptosis of T cells)	[38]
Ascites	Ultracentrifugation	Fas ligand and TRAIL	DC/PBMC	Immune suppression (apoptosis of DC/PBMC)	[39]
TAM	Kit (SBI System Biosciences)	miR-223	Cancer cell	Chemoresistance (inhibition of PTEN in cancer cell)	[40]
CAA	Ultracentrifugation	miR-21	Cancer cell	Inhibition of apoptosis (inhibition of APAF1 in cancer cell)	[41]
CAF	Ultracentrifugation	TGF-beta	Cancer cell	Migration/invasion (EMT of cancer cell)	[42]

Abbreviations: MMP-1: matrix metalloproteinase 1; ATF2: activating transcription factor 2; MTA1: metastasis associated 1; ROCK: rho-associated kinase; MALAT1: metastasis associated in lung adenocarcinoma transcript-1; ARG-1: arginase-1; PMC: peritoneal mesothelial cells; NFκB: nuclear factor-kappa B; HUVEC: human umbilical vein endothelial cell; AD-MSC: adipose tissue-derived mesenchymal stem cell; TAM; tumor associated macrophage; CAA: cancer associated adipocyte; CAF: cancer associated fibroblast; STAT3: signal transducer and activator of transcription 3; DNAM1: DNAX Accessory Molecule-1; EpCAM: epithelial cellular adhesion molecule; uPA: urokinase plasminogen activator; MT1-MMP: membrane type 1 matrix metalloproteinase; TRAIL: TNF related apoptosis inducing ligand; TGF: transforming growth factor; NK: natural killer; DC: dendritic cell; PBMC: peripheral blood mononuclear cell; APAF1: apoptotic protease activating factor 1; EMT: epithelial-to-mesenchymal transition.

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
