# Peer review of "Role of the Exosome in Ovarian Cancer Progression and Its Potential as a Therapeutic Target"

_cancers, 2019, doi:10.3390/cancers11081147_

Round 1
Reviewer 1 Report
The review on the role of exosomes in the ovarian cancer progression by Nakamura et al is a good read. The article is comprehensive, well written and sound. I do not find any major flawes. My minor suggestion to the Authors is to consider updating the 2.4.4. paragraph on the immune cells by discussing the recently published study by Czystowska et al (Nat Commun. 2019 Jul 5;10(1):3000. doi: 10.1038/s41467-019-10979-3) on the role of arginase-1-containing exosomes in the development of antitumor immune response in the ovarian cancer. Moreover, the Authors might consider adding a short passage summarizing the current knowledge on the prognostic value of exosomes in ovarian cancer and their role in the response of ovarian cancer patients to the treatment (see papers by Szajnik et al. Gynecol Obstet (Sunnyvale). 2013 Apr 29;Suppl 4:3. and Expert Rev Mol Diagn. 2016 Aug;16(8):811-26. doi: 10.1080/14737159.2016.1194758.).
Author Response
Reviewer 1
The review on the role of exosomes in the ovarian cancer progression by Nakamura et al is a good read. The article is comprehensive, well written and sound. I do not find any major flawes. My minor suggestion to the Authors is to consider updating the 2.4.4. paragraph on the immune cells by discussing the recently published study by Czystowska et al (Nat Commun. 2019 Jul 5;10(1):3000. doi: 10.1038/s41467-019-10979-3) on the role of arginase-1-containing exosomes in the development of antitumor immune response in the ovarian cancer.
We appreciate this suggestion. We added the key finding of this paper to the manuscript in Lines 260-264.
Moreover, the Authors might consider adding a short passage summarizing the current knowledge on the prognostic value of exosomes in ovarian cancer and their role in the response of ovarian cancer patients to the treatment (see papers by Szajnik et al. Gynecol Obstet (Sunnyvale). 2013 Apr 29;Suppl 4:3. and Expert Rev Mol Diagn. 2016 Aug;16(8):811-26. doi: 10.1080/14737159.2016.1194758.).
We agree with this comment. However, there are already several comprehensive reviews describing the potential of exosomes as biomarkers in ovarian cancer (e.g. J Ovarian Res. 2017; 10(1): 3., Onco Targets Ther. 2018; 11: 2831–2841.). Therefore, we aimed to differentiate our manuscript from previous reviews through focusing on the role and therapeutic potential of exosomes in ovarian cancer progression, and the current knowledge on the prognostic value of exosomes was not included in this review. We wish the reviewers to understand our thought.
Reviewer 2 Report
The paper by Nakamura et al. is a synthetic summary of the current knowledge on the role of exosomes in the dissemination of ovarian cancer. The paper has an interesting focus and a useful organization but has also some weaknesses. A major problem is that for many studies the conclusions are reported, but the methodological approach is not explained, so it is not evident for the studies reviewed whether the “involvement” in the process was demonstrated experimentally ion the study or suggested by experimental findings. As for example, page 4, line 98, it is stated that “…these exosomes had more potent oncogenic proteins, such as STAT3 and FAS, which promote cell migration and invasion…”. However, it is not clear if in this study whether STAT3 and FAS, known to promote cell migration and invasion, were simply identified in these exosomes by proteomic approach or the ability to activate Fas signalling was examined by an in vivo approach on target cells. Similar problems are present in section 2.2. Although section 2.3. and following ones are more exhaustive, the same problem is present in section 2.4.4, line 221. Details should be given for at least a few key papers in every section in order to usefully illustrate the state-of-the-art on this topic to readers, i.e. when involvement were demonstrated and when just suggested by -omic studies. These details should also include methods of exosome separation, as it is widely acknowledged that separation methods greatly affect biochemical and functional studies on exosomes (Thery et al., JEV 2018). The use of the term exosomes is now questioned and should be explained (Kowal et al., PNAS 2016; Thery et al., JEV 2018). Moreover, some details should be also added to introductory sentence to give a more complete picture, as for example at page 5, line 110 the sentence “…which is enriched with tumor-promoting signals that support ovarian cancer cell proliferation…” should list these tumor-promoting signals.
Minor points:
Table 1 is very long and not very clear, as it is not evident what was the underlying classification criteria. Perhaps it could be organized in two tables based on the source of exosomes. i.e. cancer cells or other sources.
In Figure 1, there is a discrepancy between what is graphically shown, as the picture suggests detachment from the ovary, whereas the text also indicates the tube.
The abbreviation PMC is important for the Table and the picture, but it is only given in the section 2.3. It should be given also in the caption of Figure 1 and in the Legend of Table 1
Author Response
Reviwer2
The paper by Nakamura et al. is a synthetic summary of the current knowledge on the role of exosomes in the dissemination of ovarian cancer. The paper has an interesting focus and a useful organization but has also some weaknesses. A major problem is that for many studies the conclusions are reported, but the methodological approach is not explained, so it is not evident for the studies reviewed whether the “involvement” in the process was demonstrated experimentally ion the study or suggested by experimental findings. As for example, page 4, line 98, it is stated that “…these exosomes had more potent oncogenic proteins, such as STAT3 and FAS, which promote cell migration and invasion…”. However, it is not clear if in this study whether STAT3 and FAS, known to promote cell migration and invasion, were simply identified in these exosomes by proteomic approach or the ability to activate Fas signalling was examined by an in vivo approach on target cells. Similar problems are present in section 2.2. Although section 2.3. and following ones are more exhaustive, the same problem is present in section 2.4.4, line 221. Details should be given for at least a few key papers in every section in order to usefully illustrate the state-of-the-art on this topic to readers, i.e. when involvement were demonstrated and when just suggested by -omic studies. These details should also include methods of exosome separation, as it is widely acknowledged that separation methods greatly affect biochemical and functional studies on exosomes (Thery et al., JEV 2018).
We appreciate these valuable suggestions. We extensively revised the manuscript in order to clarify what types of experiments were done in each study. We understand that the separation method of exosomes is very important. Since it is too busy to include each method in the manuscript, the information of separation methods is included in the revised Table 1.
The use of the term exosomes is now questioned and should be explained (Kowal et al., PNAS 2016; Thery et al., JEV 2018).
As pointed out, we explained that exosome is nowadays considered as a specific population of extracellular vesicles in the revised version. Please see Line 33-35.
Moreover, some details should be also added to introductory sentence to give a more complete picture, as for example at page 5, line 110 the sentence “…which is enriched with tumor-promoting signals that support ovarian cancer cell proliferation…” should list these tumor-promoting signals.
As suggested, we added several sentences to explain in detail. Please see lines 133-136.
Minor points:
Table 1 is very long and not very clear, as it is not evident what was the underlying classification criteria. Perhaps it could be organized in two tables based on the source of exosomes. i.e. cancer cells or other sources.
With this issue, I would like to follow the direction from the editor. We feel it is not practically difficult to separate this table into two because ascites should include cancer cells.
In Figure 1, there is a discrepancy between what is graphically shown, as the picture suggests detachment from the ovary, whereas the text also indicates the tube.
In Figure 1, we drew both the ovary and the fallopian tube as the primary site. As we mentioned in the text, it is now believed that some of high grade serous ovarian cancers are derived from fallopian tube. However, other histotypes, such as endometrioid, clear cell, and mucinous ovarian cancers are originated from ovary. Therefore, we would like to include both of ovary and fallopian tube as a primary site in this figure.
The abbreviation PMC is important for the Table and the picture, but it is only given in the section 2.3. It should be given also in the caption of Figure 1 and in the Legend of Table 1
According to the suggestion, we add the description in table 1. In figure 1, this abbreviation was not included.
Reviewer 3 Report
Dear Authors
The idea to summarize the „Role of the exosome in ovarian cancer progression and its potential as a therapeutic target” may be a very good chance to present all information about this topic.
However, I have some little comments to this manuscript:
I missed the references in line 42 and in line 208.
In Table 1, the abbreviations should have been extended, because I missed some explanations of abbreviations: APAF-1, NK, CAF, NFκB, FAS, STAT3, NKG2D, DNAM-1, EpCAM, MMP, TGF, and TRAIL. According to my opinion, the abbreviations should place before the References, and add other abbreviations from the text.
In Table 1 in line of TAM, please, correct the “innibition” to “inhibition”.
Please, use italics in case of the title of 2.2.1 and 2.2.2.
Line 108: “Studies” instead of “studies”.
Line 474: the journal name in italics and the year in bold.
You should use sharper figures, because after printing, the texts from the figures do not read well.
Author Response
The idea to summarize the „Role of the exosome in ovarian cancer progression and its potential as a therapeutic target” may be a very good chance to present all information about this topic. However, I have some little comments to this manuscript:
I missed the references in line 42 and in line 208.
As pointed out, we added the references in revised manuscript.
In Table 1, the abbreviations should have been extended, because I missed some explanations of abbreviations: APAF-1, NK, CAF, NFκB, FAS, STAT3, NKG2D, DNAM-1, EpCAM, MMP, TGF, and TRAIL. According to my opinion, the abbreviations should place before the References, and add other abbreviations from the text.
Thank you for the suggestion, we added these abbreviations in revised table1.
In Table 1 in line of TAM, please, correct the “innibition” to “inhibition”.
Please, use italics in case of the title of 2.2.1 and 2.2.2.
Line 108: “Studies” instead of “studies”.
Line 474: the journal name in italics and the year in bold.
Thank you for pointing them out. We corrected grammatical mistakes.
You should use sharper figures, because after printing, the texts from the figures do not read well.
We replaced the figures by the sharper imaged version in the revised version.
Round 2
Reviewer 2 Report
The prevised paper by Nakamura et al. has improved the description of the several individual studies listed, making the illustration of the current knowledge on the topic easier to catch
A few points:
Please check The English use in the sentence “Exosome is a specific population of small EVs. They are derived from endosome, and 30-150 nm-sized EVs that play key roles in cancer biology…”
Table 1 layout is still not very clear and easy to catch. Please try to improve the layout
Author Response
The prevised paper by Nakamura et al. has improved the description of the several individual studies listed, making the illustration of the current knowledge on the topic easier to catch.
Please check The English use in the sentence “Exosome is a specific population of small EVs. They are derived from endosome, and 30-150 nm-sized EVs that play key roles in cancer biology…”
As the reviewer pointed out, we checked and revised the sentence. Plsease see line 34-38.
Table 1 layout is still not very clear and easy to catch. Please try to improve the layout
As suggested, we divided the table into two parts depending on the source of exosomes.